# Biological Traits of the Pincer Wasp *Gonatopus flavifemur* (Esaki & Hashimoto) Associated with Different Stages of Its Host, the Brown Planthopper, *Nilaparvata lugens* (Stål)

**DOI:** 10.3390/insects11050279

**Published:** 2020-05-02

**Authors:** Jiachun He, Yuting He, Fengxiang Lai, Xiangsheng Chen, Qiang Fu

**Affiliations:** 1Institute of Entomology, Special Key Laboratory for Development and Utilization of Insect Resources of Guizhou, Guizhou University, Guiyang 550025, China; hejiachun1984@126.com; 2State Key Laboratory of Rice Biology, China National Rice Research Institute, Hangzhou 310006, China; heyuting0309@126.com (Y.H.); laifengxiang@caas.cn (F.L.); 3Hunan Provincial Key Laboratory for Biology and Control of Plant Diseases and Insect Pests, Hunan Agricultural University, Changsha 410128, China

**Keywords:** *Gonatopus flavifemur*, *Nilaparvata lugens*, host stage, parasitism, host feeding, development

## Abstract

*Gonatopus flavifemur* (Esaki & Hashimoto) is a common parasitoid of the most important rice pest, the brown planthopper (BPH) *Nilaparvata lugens* (Stål), in eastern and southeastern Asia. We investigated the parasitism rates, feeding rates, and offspring development of *G. flavifemur* in association with five instars of BPH nymphs and male and female adults under laboratory conditions (27 ± 1 °C and 70 ± 5% Relative Humidity). The results showed that the life stage of the host significantly affects parasitism, host feeding, and offspring development by *G. flavifemur*. The parasitism rate was highest on 4th instar nymphs, and the feeding rate was highest on 1st instar nymphs. The cocooning rate on male adult BPHs was significantly lower than that on other stages; however, emergence rates did not significantly differ among the BPH stages. The rate of female offspring upon emergence from 5th instars was higher than other stages. Both the parasitism and host-feeding functional responses of *G. flavifemur* to different BPH stages fit well with Holling type II models, supporting the results of parasitism and feeding rates and indicating that *G. flavifemur* would be a good agent for BPH control. In conclusion, *G. flavifemur* prefers to feed on young nymphs but prefers to parasitize older nymphs. In addition, 5th instar nymphs are favorable to female offspring of the pincer wasp.

## 1. Introduction

Dryinids are the main parasitic wasps of planthoppers in rice fields and can parasitize and feed on both nymphs and adults [1]. Approximately 12 dryinid species have been reported to parasitize rice planthoppers in China. Among them, *G. flavifemur* (Esaki & Hashimoto), *Gonatopus nigricans* (R.C.L. Perkins), *Haplogonatopus oratorius* (Westwood), *Haplogonatopus apicalis* R.C.L. Perkins, and *Echthrodelphax fairchildii* R.C.L. Perkins are reported to be common species in rice fields [2,3,4,5]. In addition, *Gonatopus flavifemur* (Esaki & Hashimoto, 1932) (synonym: *Pseudogonatopus flavifemur* Esaki & Hashimoto) is one of the most common natural enemies of rice planthoppers (Delphacidae) in China [6]. This species is consistently characterized by sexual dimorphism [7], and the wingless, ant-like female (Figure 1B) wasps lay eggs and feed on planthoppers (Figure 1C). The mode of reproduction in *G. flavifemur* is sexual and parthenogenetic. Typically, females lay eggs on a live host, and the wasp larvae hatch and grow on the host, which are usually sacciform and endo-ectoparasitic (Figure 1D). The host dies after the larvae leave the host to cocoon. Female feeding involves consumption of hemolymph and tissue, which inflicts wounds on the host body and leads to host death (Figure 1E) [8,9,10]. However, male *G. flavifemur* wasps are fully winged (Figure 1A) and do not feed on hosts or feed only on sugar solutions. The males search for females with which to mate upon emergence [11,12,13]. *G. flavifemur* was first reported to be a natural enemy of rice leafhoppers and planthoppers by Esaki and Hashimoto in 1932 [14]. Since then, there have been considerable reports about its distribution. Xu et al. and Olim et al. revised the family Dryinidae in the oriental region and eastern Palearctic, and the results showed *G. flavifemur* to be distributed in China, Japan, the Philippines, Malaysia, India, and Australia. In China, *G. flavifemur* is widely distributed in Jiangsu, Zhejiang, Hunan, Jiangxi, Anhui, and 13 other provinces [15,16]. Studies on the biological habits of dryinids show that in rice fields these wasps can parasitize several species of planthoppers; however, the main hosts differ. *G. flavifemur* can parasitize seven species of planthoppers, but the main host is the brown planthopper (BPH) *Nilaparvata lugens* (Stål) [8,11,17].

The BPH is one of the most important pests in rice fields in eastern and southeastern Asia [18]. Since the 1980s, the main method of controlling BPHs in Asian countries has been to use chemical insecticides, but the excessive and single use of the same chemical insecticide has caused resistance in BPHs to many insecticides to increase each year [19]. Moreover, a large number of natural enemies have also been killed, and these factors have led to the resurgence of rice planthoppers [20]. Therefore, the enhancement of natural control factors, protection, and rational use of natural enemies to control planthoppers have become very important [21]. For these reasons, many researchers have begun to focus on the study of the natural enemies of rice planthoppers. *G. flavifemur* is a common natural enemy of BPH, and some reports from China, Japan, and other countries have assessed its biological habits.

In 1982, Yang et al. described 10 species of dryinids in rice fields of China, including *G. flavifemur*, and studied their biological habits [11]. Huang reported that *G. flavifemur* lay eggs on BPHs, white-backed planthoppers (WBPHs), small brown planthopper (SBPHs), and *Nilaparvata bakeri* (Muir), and the larvae can develop and emerge. However, when females parasitize *Sogatella vibix* (Haupt)*, Metadelphax propinqua* (Fieber), and *Saccharosydne procerus* (Matsumura), the larvae in the host all die [8]. In examining the functional response of *G. flavifemur* to 3rd instar BPHs, Chua et al. showed that as the density of female wasps increased, the search efficiency of *G. flavifemur* for 3rd instar BPHs decreased and that the processing time of each host significantly increased [7]. In addition, Sahragard et al. found that *G. flavifemur* can parasitize nymph and adult BPHs and that the fertility of females correlated positively with the number of hosts [22]. In addition, the wasp has an obvious preference for particular hosts; under choice conditions, in which BPHs and WBPHs are introduced into the same cage, females prefer to parasitize BPHs and feed on WBPHs [23]. From 2015 to 2019, we investigated the parasitism of BPHs in Guizhou, Yunnan, Hubei, Hunan, and Zhejiang provinces in China, and the results showed that *G. flavifemur* is the main parasitic wasp of nymph and adult BPHs in the rice fields of southern China, with a parasitism rate of 1.06–7.29% (unpublished data). These findings show that *G. flavifemur* is one of the most common parasitoids of BPHs in southern China.

To date, there have been several preliminary reports on *G. flavifemur* exploring its habits given that it is considered one of the most common natural enemies in rice paddies. However, there has been no systematic assessment of its ability to control different stages of BPHs. In this study, we examined the efficacy of *G. flavifemur* against BPHs and evaluated the effect of host stage on the development of *G. flavifemur* F1 offspring.

## 2. Materials and Methods

### 2.1. Materials and Experimental Conditions

Brown planthoppers were collected from rice fields in Fuyang district, Hangzhou city, Zhejiang Province, China, and reared on a susceptible rice variety (Taichung Native 1, TN1) in the laboratory (27 ± 1 °C, 70 ± 5% Relative Humidity) over multiple generations. The parasitoid *G. flavifemur* was also collected from the above BPHs, whose bodies showed sac protrusion indicating parasitization. The wasps were reared on 4–5th instar BPHs under the same conditions.

The rice plants were transplanted into seedling pots (90 mm diameter), which were cleaned and contained 2–3 tillers. The plants were covered with a cylindrical plastic cage, 75 mm in diameter and 250 mm in height. The opening at the upper end of the cage was covered with gauze, which allowed for easy introduction of BPHs and *G. flavifemur*. The gauze cover provided ventilation and prevented the test insects from escaping and entering other cages. The rice plants were used in the experiments one week after transplanting.

All experiments were conducted in a greenhouse constructed of iron and glass using a constant temperature and humidity control system (WS-SL-1600S) that maintained temperature at 27 ± 1 °C and humidity at 70 ± 5% Relative Humidity under natural light (July 2019, average daylength about 14 h).

### 2.2. Effects of Different Stages of BPHs on Parasitism, Host Feeding, and Offspring Development in Gonatopus flavifemur

The experiment assessed all 7 stages of BPHs, which have 5 instars of nymphs—1st, 2nd, 3rd, 4th and 5th—and male and female adults [21,24]. Thirty BPHs of each stage were introduced into a cage and allowed to settle on the plant inside for approximately 1 h, leaving a pair of newly emerged female and male *G. flavifemur* mates, after which only the female was released. There were 9–10 replicates for each stage.

After 24 h, the *G. flavifemur* individuals were removed, and the number of BPH hosts fed upon by *G. flavifemur* was counted by looking for signs such as wounds on the host body (Figure 1D) under the microscope. Living BPHs were observed daily until their bodies showed sac protrusion (older than 2nd instar larvae, Figure 1E). The number of parasitized hosts and larval sacs per host were counted. The rate of parasitism, host feeding, and superparasitization (the proportion of hosts with more than 1 larval sac on its body) and the number of larval sacs were quantified. Parasitized BPHs were checked daily, and the onset of larval cocooning (Figure 1F) and days to emergence were recorded. The numbers of cocoon and male and female offspring that emerged from the cocoons were counted, and the stages and numbers of dead hosts were recorded. Then, the developmental time, cocooning, and emergence rates of the wasp and percentage of female offspring and proportions of the different host stages reached (%) at the time of parasitoid cocooning were estimated.

### 2.3. Parasitism and Host-Feeding Functional Response of the Parasitoid Gonatopus flavifemur Associated with Different Stages of BPHs

The experiment involved 4 stages of BPH: 1st–2nd instar nymphs, 4th–5th instar nymphs, and male and female adults. The host density at each stage was 2, 4, 8, 16, 32, 64, or 128 BPHs per plant, with 3 replicates for each host density and stage combination. This test was carried out as described above, but we counted only the number of BPHs parasitized and fed upon by *G. flavifemur*.

### 2.4. Data Analyses

All bioassay data were analyzed by using one-way ANOVA and Tukey’s multiple comparison tests. Before ANOVA, normality of the data was checked, percentage data were arcsine transformed, and untransformed means (±SE) presented. All data were tested by the Bartlett Chi-squared test method and met the assumption of homogeneity of variance. All statistical calculations were performed with Data Processing System (DPS) version 15.1 software [25].

Functional response analysis was implemented by using a Holling type II functional response model [26]:$$\mathrm{Na}\;=\frac{\mathrm{aTN}}{1+\mathrm{aThN}}$$
$${\mathrm{Na}}_{max}=\frac{1}{\mathrm{Th}}$$
where N is the host density, Na is the number of host attacks (hosts were parasitized/fed), a is the attack rate, T is the time available for searching (total experimental period was 1 day), Th is the handling time (i.e., the time taken for a wasp to quell and parasitize/feed on a host, clean itself and rest after parasitizing/feeding on a host), and Na_max_ is the maximum number of hosts parasitized/fed upon daily (1 d). All functional response parameters were calculated using a nonlinear regression model with DPS software.

## 3. Results

### 3.1. Effects of Different Stages of BPHs on Parasitism and Host-Feeding Rates in Gonatopus flavifemur

Parasitism and host-feeding rates of *G. flavifemur* were significantly different between the life stages of BPHs (Tukey’s multiple comparison tests: parasitism: *F* = 11.52, *df* = 6/58, *p* < 0.001; host feeding: *F* = 39.69, *df* = 6/58, *p* < 0.001) (Figure 2A,B).

The rate of parasitism on 4th instar nymphs (58.52 ± 5.69%) was the highest among the tested stages, followed by 3rd (52.33 ± 3.74%), 2nd (41.48 ± 5.52%), and 5th (38.33 ± 5.17%) instars, but no significant difference was detected among these four stages. However, the rate of parasitism on 4th instars was significantly higher than that on female (36.67 ± 2.94%) and male (20.33 ± 1.79%) adults and 1st instars (17.41 ± 4.41%). The lowest parasitism rate was found on 1st instars (17.41 ± 4.41%).

The feeding rate on 1st instars nymphs (38.52 ± 3.19%) was the highest and significantly higher than that on the other stages of BPH, followed by 2nd (22.96 ± 1.85%), 3rd (19.67 ± 1.29%), 4th (12.22 ± 1.17%), and 5th (12.08 ± 1.17%) instars and male (7.00 ± 0.99%) and female (5.67 ± 0.82%) adults. The feeding rates on 2nd and 3rd instars were significantly higher than those on 4th and 5th instars as well as on male and female adults. The feeding rate on female adults was the lowest and significantly lower than that on all nymph stages.

The percentage of superparasitization among the seven stages of BPH did not significantly differ (Tukey’s multiple comparison tests: *F* = 2.06, *df* = 6/58, *p* = 0.072) (Figure 2C). The percentage of superparasitization on 2nd instars (19.66 ± 7.11%) was the highest, followed by 3rd instars (16.14 ± 5.96%), 4th instars (11.08 ± 6.72%), females (9.78 ± 2.73%), 5th instars (6.69 ± 4.87%), males (3.67 ± 2.46%), and 1st instars (2.98 ± 2.15%).

The results regarding the number of larval sacs among the tested individuals also differed significantly (Tukey’s multiple comparison tests: *F* = 7.05, *df* = 6/58, *p* < 0.001) (Figure 2D). Similar to the parasitism rate results, the number of larval sacs was the highest on 4th instar nymphs (20.4 ± 3.5), followed by the 3rd (18.9 ± 2.6), 2nd (15.0 ± 2.3), and 5th (12.8 ± 2.6) instar nymphs and female (12.3 ± 1.3) adults, but there were no significant differences among these five host stages. Male (6.3 ± 0.6) and 1st instar (5.4 ± 1.5) hosts showed significantly lower numbers than 4th and 3rd instar hosts.

### 3.2. Effects of Different Stages of BPHs on Gonatopus flavifemur Offspring Development

The cocooning rate of *G. flavifemur* offspring when parasitizing male adults (39.15 ± 11.64%) was significantly lower than that when parasitizing other stages (Tukey’s multiple comparison tests: *F* = 3.73, *df* = 6/57, *p* = 0.003) (Figure 3A). The cocooning rate on 2nd instars (84.98 ± 4.52%) was the highest, followed by 4th (79.87 ± 7.38%), 3rd (78.73 ± 2.65%), 1st (71.01 ± 6.97%), and 5th (71.00 ± 4.29%) instars and female adults (70.56 ± 6.29%). There were no significant differences among these stages. Additionally, no significant difference among all stages was observed for the emergence rate (Tukey’s multiple comparison tests: *F* = 0.48, *df* = 6/55, *p* = 0.829) (Figure 3B). The emergence rate was the highest in 2nd (91.52 ± 2.76%) instars, followed by 4th instars (91.49 ± 3.88%), 5th instars (91.11 ± 4.46%), 3rd instars (90.74 ± 3.51%), female adults (87.37 ± 3.49%), 1st instars (85.00 ± 11.59%), and male adults (71.88 ± 14.95%).

Among the offspring, the number and percentage of females differed significantly among the seven stages of host development (Tukey’s multiple comparison tests: number: *F* = 6.85, *df* = 6/53, *p* = 0.000, percentage: *F* = 13.28, *df* = 6/53, *p* = 0.000) (Figure 3C,D). The highest number and percentage of females were found for 5th instar hosts (number: 11.6 ± 0.7, percentage: 57.25 ± 6.13%). However, there was no significant difference in the number and percentage of females between the 5th and 4th instar (number: 11.1 ± 1.4, percentage: 27.90 ± 4.11%) and female adult (number: 9.5 ± 1.7, percentage: 40.00 ± 9.02%) hosts. The number of females in the 3rd instar (7.8 ± 1.6) hosts was not significantly different from that in the 5th instar hosts, but the percentage of females in 3rd instar (17.13 ± 3.46%) hosts was significantly lower than that in the 5th instar hosts. These two parameters for adult males (number: 1.4 ± 1.4, percentage: 11.11 ± 10.14%) and 2nd (number: 1.3 ± 0.8, percentage: 1.75 ± 1.11%) and 1st (number: 0, percentage: 0%) instar hosts were all significantly lower than those for 5th instar hosts. In particular, no females emerged from hosts in the 1st instar stage.

The developmental time from egg to cocoon significantly differed (Tukey’s multiple comparison tests: *F* = 17.63, *df* = 6/55, *p* = 0.000) (Figure 4A). Indeed, the time for egg to cocoon in 3rd (10.0 ± 0.1 d) and 4th (9.9 ± 0.1 d) instars was significantly longer than that in other stages, with no significant difference among the other stages (1st: 9.2 ± 0.1 d, 2nd: 9.2 ± 0.2 d, 5th: 9.1 ± 0.1 d, male: 8.8 ± 0.1 d, female: 8.7 ± 0.1 d). The developmental time from cocoon to adult also differed significantly among the seven host stages (Tukey’s multiple comparison tests: *F* = 23.69, *df* = 6/52, *p* = 0.000) (Figure 4B). The time from cocoon to adult in the 3rd (12.6 ± 0.1 d) instar was significantly longer than that in other stages, followed by the 5th instar (12.1±0.2 d), which was also significantly longer than the other stages (1st: 11.0 ± 0.2 d, 2nd: 11.1 ± 0.1 d, 4th: 11.4 ± 0.2 d, male: 11.1 ± 0.1 d, female: 11.2 ± 0.1 d). There were no significant differences among the five other stages. The developmental times of males and females from egg to adult were significantly different among all host stages (Tukey’s multiple comparison tests: male: *F* = 72.50, *df* = 6/406, *p* = 0.000; female: *F* = 27.73, *df* = 5/137, *p* = 0.000). The developmental time of males in the 3rd instar (22.5 ± 0.1 d) stage was significantly longer than that of males in the other stages, and that of males in the 4th (20.8 ± 0.1 d) and 5th (21.0 ± 0.2 d) instar stages was significant longer than that of males in the four other host stages (1st: 20.2 ± 0.2 d, 2nd: 20.2 ± 0.1 d, male: 19.7 ± 0.2 d, female: 19.9 ± 0.1 d), and there was no significant difference among the other stages (Figure 4C,D). The developmental time of females in the 3rd (22.6 ± 0.2 d) and 4th (22.0 ± 0.2 d) instars was significantly longer than that of females in the 5th (21.2 ± 0.1 d) instar and of female (19.9 ± 0.2 d) and male (20.5 ± 1.5 d) adults, and there were no significant differences among the last three stages. In addition, there was no significant difference between the 2nd (21.5 ± 0.5 d) instar stage and all other host stages.

We found that the majority of parasitized BPH nymphs could grow and even molt. Moreover, most parasitized young nymphs (>94%) could molt twice; 100% of 1st instars were able to survive through to the 3rd instar after being parasitized, 94% of 2nd instars survived through to the 4th instar, and 96.58% of 3rd instars survived through to the 5th instar. A few 2nd (5.6%) and 3rd (3.4%) instar nymphs were able to molt three times and survive through to the 5th instar and adult stages. Approximately 80.6% of 4th instars could molt one time to the 5th instar stage, 19.4% of 4th instars could molt twice and emerge, and all 5th instar hosts could emerge (Table 1).

### 3.3. Parasitism and Host-Feeding Functional Response

The parasitism and host-feeding functional response of *G. flavifemur* to different stages of BPHs fit well with the Holling type II models (R^2^ >95%) (Figure 5, Table 2). The functional response curves of the four stages of the host showed that the number of hosts parasitized/fed upon increased steeply at low-host densities and then tended to flatten and finally reached an asymptote at high-host densities.

The estimated values of the attack rates and handling times associated with the four host stages are shown in Table 2. The attack rates (a) of parasitism were found to be the highest for 4th–5th instars (0.864), followed by female and male adults and 1st–2nd instars. The handling time (Th) associated with the parasitization of 4th–5th instars was the shortest (0.036), followed by 1st–2nd instars and female and male adults. The maximum daily parasitization rate (Na_max_) was the highest for 4th–5th instars (27.579), followed by 1st–2nd instars and female and male adults.

The attack rates (a) associated with host feeding were found to be the highest for 1st–2nd instars (0.860), followed by 4th–5th instars and male and female adults, and the handling time (Th) associated with host feeding was the shortest for 1st–2nd instars (0.036), followed by the 4th–5th instars and male and female adults. The maximum daily rate of host feeding (Na_max_) was also observed for 1st–2nd instars (24.882), followed by 4th–5th instars and male and female adults.

## 4. Discussion

Females of most species of dryinids can parasitize and feed on hosts [13]. Host stage has a very strong influence on the development of *Gonatopus flavifemur*. The results of this study showed that *G. flavifemur* can parasitize and feed upon all life stages of BPH. However, marked differences were observed in the parasitism rate, host-feeding rate, and development of offspring when females were confronted with different BPH stages.

The parasitism rate was highest in 4th instars; as the age of the BPHs increased or decreased, the parasitism rate gradually decreased. Based upon our results for the parasitism functional response, attack rates were highest against 4th–5th instars, with the handling time being the shortest. Attack rates and handling time are the most important parameters in evaluating the functional response [26]. The handling time reflects the consumption rate and effectiveness of a predator, whereby a short handling time indicates a relatively high consumption rate and effectiveness [27]. Therefore, the parasitism rate was highest on 4th instars because the effectiveness of parasitizing 4th–5th instars was higher than that of parasitizing the other instars. When the age of BPHs increases or decreases, the attack rate decreased, and the handling time increased, causing the parasitism rate to decrease. Our findings on the parasitism rate of hosts of different life stages agree with the “dome-shaped” hypothesis [28], which states that the intermediate age of host larvae/nymphs is richer in quality over the two extremes of the host development stage with regard to the fitness of parasitic wasps in terms of laying eggs. Several studies have documented that defense ability against parasitoids and the size of hosts are the main factors affecting host-selection behavior in parasitic wasps [29,30]. In general, the defense ability of a host refers to its escape and resistance strength against capture by parasitoids. This defense ability increases with age, and thus, relatively old hosts are difficult to subdue [31]. Nonetheless, the size of a host is an important standard by which to measure host quality: relatively old nymph or adult hosts usually contain more resources and yield greater production of large parasitoid offspring [32,33]. Parasitic wasps must weigh host defense ability and quality to select the best host [34,35]. Our results show that the parasitic behaviors of *G. flavifemur* fit well with the “dome-shaped” hypothesis.

According to our results, the feeding rate was significantly higher on 1st instars than on other instars. The feeding rate decreased with BPH age. In addition, the host-feeding functional response results showed the highest attack rates for 1st–2nd instars, and the handling time was the shortest. The attack rates decreased, and the handling time increased, with age. These results indicate that feeding effectiveness decreases as the age of BPH increases, and thus, the feeding rate decreases. Heimpel et al. reported that host feeding and oviposition are strongly influenced by host age or quality. Young hosts are small, their quality is poor, and they easily die after being parasitized; thus, they are more likely to be prey than to lay eggs [36,37]. As younger hosts contain fewer resources, parasitoids need to feed on more hosts to obtain sufficient resources. Therefore, *G. flavifemur* may prefer to feed upon younger and more easily caught hosts and may need to feed on a larger number of individuals to obtain the same resources as they would with older hosts. Many scholars believe that the behavior of feeding on hosts is an advantage of the use of parasitoids for pest control [38]. Comparing the parasitism rate and feeding rate, we found that the parasitism rate on 1st instar nymphs was lowest but that the feeding rate on this stage was highest among the instars. Therefore, host feeding can improve the control effect of parasitic wasps on different stages of the host.

Additionally, the parasitism and host-feeding functional response of *G. flavifemur* to different stages of BPHs fit the Holling type II model well. Using a functional response model to analyze the parasitic and host-feeding function of natural enemies is an important method for systematic evaluation [39]. Our results for this species regarding the type of functional response differ from those obtained by Chua et al.); in that study, different host and parasitic wasp densities were used for experiments, and the functional response of type III was suitable. However, in our study, each treatment included only one female wasp, and our data fit the type II model. According to this model, the estimated values of attack rates, handling times, and maximum parasitization/host-feeding rates strongly support the results for parasitism and feeding rates. Our study was a laboratory test, it is not known whether the results are applicable to the field. Under natural conditions, parasitism rates are rather lower than 20% [7]. Chemical insecticides [20] and natural enemies such as hyper-parasitoids [13,40] all affect the population size and control effect of pincer wasps in the field. Even so, this study contributes to a better understanding of the biological traits of *G. flavifemur*, providing an important theoretical basis for using *G. flavifemur* to control BPHs.

The results of the effects of host stage on offspring development showed that the highest number of larval sacs occurred in 4th instars; however, the highest cocooning rate occurred in 2nd instars, and the cocooning rate decreased in older nymph and adult hosts. This result was consistent with those of Li et al. [41], who reported that as the age of the host (WBPH) increased, the survival rate of *Haplogonatopus apicalis* larvae decreased, indicating that relatively old WBPH nymphs showed a stronger defense against *H. apicalis.* Previous studies have indicated that the host’s defense ability against parasitic wasps is not only due to behavioral resistance but that there are many resistance substances secreted by a parasitized host that are detrimental to the development of parasitoids. Resistance substances can increase with the age of a host [42,43,44], which may explain our results. Furthermore, the emergence rate of BPH did not significantly differ among all developmental stages, which shows that the host had no effect on the emergence rate after the larvae of *G. flavifemur* left their host to pupate.

In this study, the developmental time of parasitoid eggs to cocoons when parasitizing 3rd instar to adult hosts decreased and that of females parasitizing 5th instar nymphs and female and male adults was significantly shorter than that of females parasitizing 3rd and 4th instar nymphs. Sequeira and Mackauer reported that the nutritional source of parasitoid growth depends on the quality of the host [45]. However, the quality of and available resources in young hosts are relatively low, and thus, a parasitic wasp must extend its development time to achieve sufficient growth [35,46]. Relatively old BPHs are larger and of better quality than younger BPHs, and parasitoids will grow faster on older hosts. We also found that the rates of female offspring produced when developing on 5th instars and female adults were higher than those produced when developing on other stages. In general, the female ratio is one of the most important factors affecting the reproduction of wasps [47], and parasitic wasps always give birth to a higher percentage of female offspring on larger or better-quality hosts [48,49]. BPHs in the 5th instar and females are larger and have better quality than other stages. Although there were no significant differences in the parasitism, cocooning or emergence rates or development time of female offspring between these two stages of hosts, the number and proportion of female offspring developing on 5th instars were higher than those on female adults. It can be inferred that 5th instars may be favorable to the growth of female parasitoids.

In addition, superparasitism is common among parasitic wasps. For dryinids, this rare phenomenon under natural conditions is often observed in the laboratory [7]. Yamada et al. reported that over 24 h, self-superparasitism behavior in *Haplogonatopus oratorius* and *Echthrodelphax fairchildii* on SBPHs did not affect the survival rate of offspring [50,51]. Their results were similar to ours, showing that the percentage of superparasitization and emergence rates among the seven stages of BPHs did not differ significantly. This result indicates that in laboratory rearing of *G. flavifemur,* superparasitism might occur at all BPH stages; over 24 h, there was no effect on the development of offspring. Although this experiment did not consider the effects of practical factors in the field, it simulated the laboratory rearing conditions of *G. flavifemur*. Therefore, the results can guide the mass rearing of *G. flavifemur*; specifically, controlling the proportions of parasitoids and hosts can ensure the effective utilization of hosts. Furthermore, it can reduce feeding costs and allow a stable parasitism rate to be maintained to ensure the quality of wasp rearing.

Interestingly, some of the 3rd–5th instar hosts could emerge as adults. Mita et al. reported that females of the genus *Haplogonatopus* are wingless, that the migration ability of adult wasps should be highly restricted, but that the passive dispersal of larvae parasitizing hosts may be possible [52]. From 2012 to 2017, we investigated BPHs collected under a light trap in more than 10 regions of southern China, and the results showed that 0.19%–2.24% of the BPHs were parasitized by dryinids (unpublished data). We can infer that if BPHs carry the larvae of dryinids to facilitate long-distance migration, the BPHs may be adults and 5th instars when they are parasitized, with some being 3rd–4th instars. Nevertheless, parasitized 1st–2nd instars are not able to carry the larvae of dryinids during long-distance migration. This result is very important for studying the migratory behavior of dryinids in the future and for the development of pest control strategies.

## 5. Conclusions

Studies have indicated that *Gonatopus flavifemur* can parasitize and feed on all life stages of BPH and is a good agent for BPH control. *G. flavifemur* prefers to feed on young nymphs of BPH, whereas older nymphs are the best host for parasitization and breeding. Additionally, 5th instar nymphs are favorable to female offspring of *G. flavifemur*.

## Figures and Tables

**Figure 1 insects-11-00279-f001:**
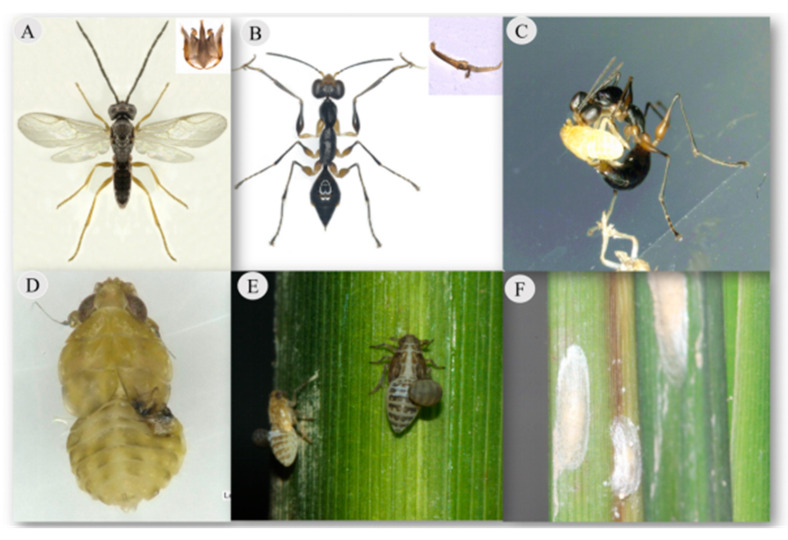
*Gonatopus flavifemur* (Esaki & Hashimoto, 1932). (**A**) Male and genitalia. (**B**) Female and chela. (**C**) Female with a brown planthopper (BPH) nymph. (**D**) Nymph after being fed upon by *G. flavifemur*. (**E**) Larval sac on a BPH nymph. (**F**) *G. flavifemur* cocoon.

**Figure 2 insects-11-00279-f002:**
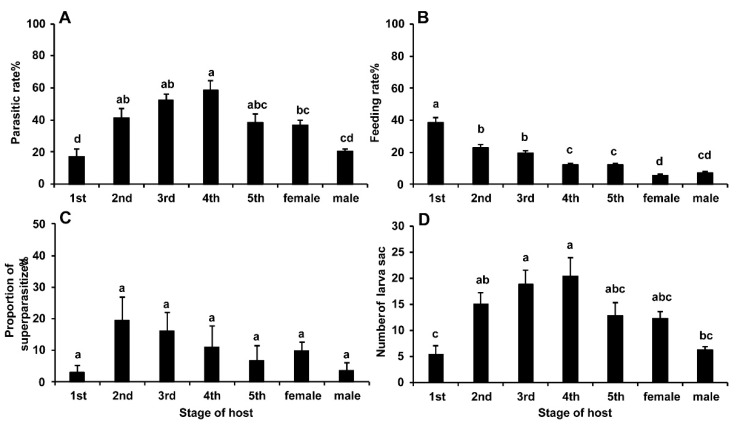
Effects of BPH life stage on parasitism and host-feeding rates in *Gonatopus flavifemur*. (**A**) parasitism rate; (**B**) feeding rate; (**C**) percentage of superparasitized individuals; (**D**) number of larval sacs. Note: bars within each panel with the same letter above show no significant difference at the 0.05 level according to Tukey’s multiple comparison tests. The same in the following figures.

**Figure 3 insects-11-00279-f003:**
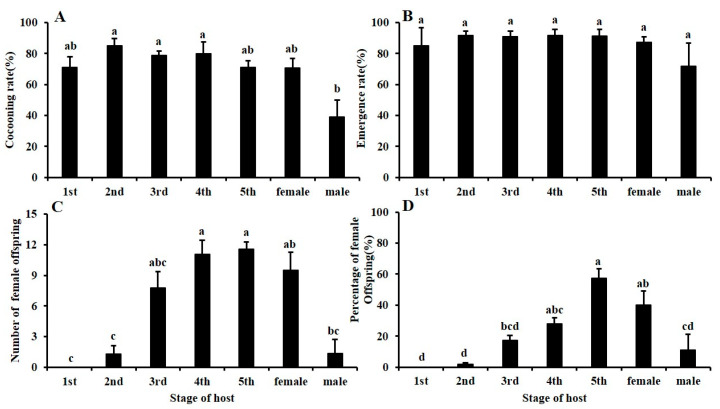
Effects of BPH life stage parasitism on (**A**) cocooning rate, (**B**) emergence rate, (**C**) number of female offspring, and (**D**) percentage of female offspring of *G. flavifemur*.

**Figure 4 insects-11-00279-f004:**
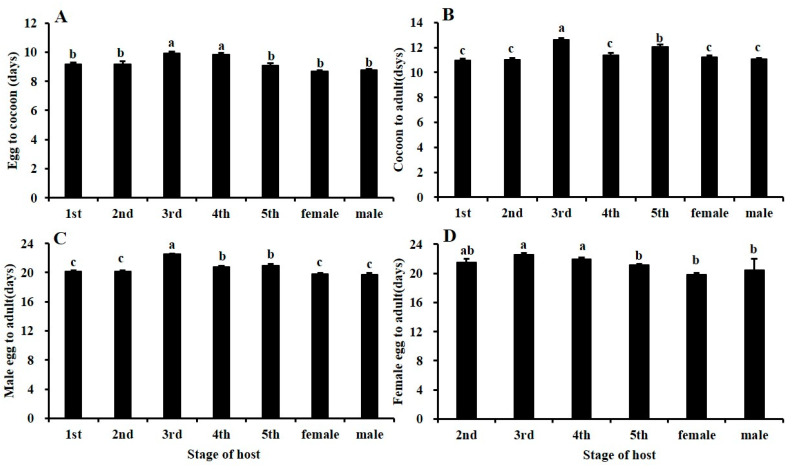
Effects of BPH life stage on the time required for the development of *Gonatopus flavifemur*: (**A**) egg to cocoon, (**B**) cocoon to adult, (**C**) egg to adult (male), and (**D**) egg to adult (female).

**Figure 5 insects-11-00279-f005:**
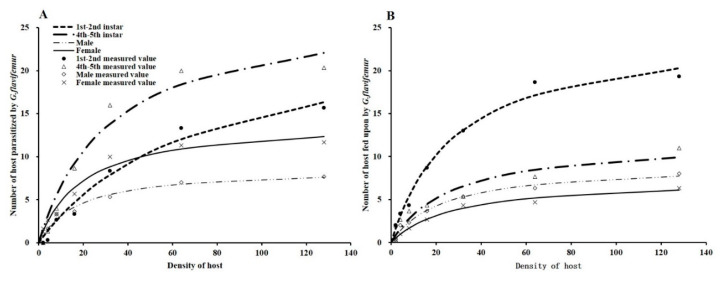
Functional response curves for *Gonatopus flavifemur* parasitism and host feeding at different stages of BPH (Holling type II). (**A**) Parasitism functional response curves. (**B**) Host-feeding functional response curves.

**Table 1 insects-11-00279-t001:** Percentages of host stages at the time of *Gonatopus flavifemur* cocooning.

Host Stage When Parasitized	Host Stage Reached (%) at Time of Parasitoid Cocooning					
	1st Instar	2nd Instar	3rd Instar	4th Instar	5th Instar	Adult
1st instar	0	0	100.00	0	0	0
2nd instar	-	0	0	94.44	5.56	0
3rd instar	-	-	0	0	96.58	3.42
4th instar	-	-	-	0	80.58	19.42
5th instar	-	-	-	-	0	100.00

**Table 2 insects-11-00279-t002:** Functional response and parameters of *Gonatopus flavifemur* parasitism and host feeding at different BPH stages (Holling type II).

Behavior	Stage of Host	Functional Response Equation	R^2^	Attack Rate (a)	Handling Time (Th/day)	Maximum Parasitism/Host Feeding (Na_max_)
Parasitism	1st–2nd instar	Na = 0.357N/(1 + 0.014N)	0.976	0.357	0.039	25.491
	4th–5th instar	Na = 0.864N/(1 + 0.031N)	0.959	0.864	0.036	27.579
	Male	Na = 0.502N/(1 + 0.057N)	0.981	0.500	0.116	8.636
	Female	Na = 0.726N/(1 + 0.051N)	0.970	0.726	0.070	14.229
Host-feeding	1st–2nd instar	Na = 0.860N/(1 + 0.035N)	0.985	0.860	0.040	24.882
	4th–5th instar	Na = 0.449N/(1 + 0.037N)	0.968	0.449	0.083	12.039
	Male	Na = 0.397N/(1 + 0.044N)	0.982	0.397	0.110	9.117
	Female	Na = 0.267N/(1 + 0.036N)	0.983	0.267	0.135	7.431

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
