# Peer review of "Biological Traits of the Pincer Wasp Gonatopus flavifemur (Esaki & Hashimoto) Associated with Different Stages of Its Host, the Brown Planthopper, Nilaparvata lugens (Stål)"

_insects, 2020, doi:10.3390/insects11050279_

Round 1
Reviewer 1 Report
Review of Jiachun He et al.
Title: Biological traits of the pincer wasp Gonatopus flavifemur (Esaki et Hashimoto) associated with different stages of its host, the brown planthopper, Nilaparvata lugens (Stål)
The authors tested the suitability of the wasp Gonatopus flavifemur as biological control agent of the planthopper, Nilaparvata lugens, the most important rice pest in eastern and southeastern Asia. They measured several G. flavifemur parameters such as parasitism rate, feeding rate as well as offspring development on five nymphal instars as well as on male and female adults under controlled conditions ( 27±1°C and 70±5% R.H.).
Results indicate that parasitism was higher in late instar, while first instars are best suited for direct feeding and that later instars are favourable to female offspring.
Both the parasitism and host-feeding functional responses of G. flavifemur to different stages of the BPH fit well with Holling type II models.
Overall, the manuscript describes the work in sufficient detail. The writing is mostly clear; however, I did find a few areas where the presentation could be improved, and will explain my questions and concerns below:
After reading the paper several questions raised in my mind, first can you discuss why your results for the same species regarding the type of functional response differ from those obtained by Chua et al. (1984)?
Second if parasitized hosts in several instars do not die, how can you strongly say or guarantee that these are effective parasitoids of the planthopper? Do these continue to feed? You refer to other Dryinid apterous parasitoid species which hijack the planthopper to disperse, meaning that the host fitness is not seriously compromised.
Further, if in the field this is one of the most abundant parasitoids, how do you explain the fact that these are not effectively controlling the pest.
If females eat first host instars, they tend to increase their fitness however, this means that later one there would be an increase in the competition for suitable host for oviposition. How does this resource competition can influence parasitism?.
What practices of biological control do you propose for this species to be used in the field?
More specific comments:
Section Introduction
It would be informative to speak a bit about the ecology of the parasitoid. For instance, photos are showed in the following sections it would be informative to refer to the fact that immature larval instars are usually sacciform and endo-ectoparasitic.
Pag 1,L 36 - Please reformulate the sentence since you are stating the obvious, no male parasitoid parasitizes any animal, this is exclusive to females.
Further, within the Dryinidae family males do not feed or feed only on sugar solutions. If you have any specific reference to prove this or if you observe males feeding on planthoppers please refer that.
Methods
Pag 3 L101 – Dryinidae reproduction may be sexual and/or parthenogenetic, how did you guaranteed that these would mate. Would not be more accurate to leave male and female mate and then release only the female?
Pag 3 L119 – Did you check for normality of the data before ANOVA? Which transformation of the data did you used?
Results
From Pag 4 L139-140, 153, 168, 169, 177, 180, 184,185 – p values of 0.000 are usually presented as p<0.001, since 0.000 can suggest that there was absolutely no (zero) chance of getting these results if the null hypothesis was true.
Author Response
Dear Reviewer:
Thank you very much for your letter and the comments from the referees about our paper submitted to Biological traits of the pincer wasp Gonatopus flavifemur (Esaki et Hashimoto) associated with different stages of its host, the brown planthopper, Nilaparvata lugens (Stål) (ID: insects-770681). We have checked the manuscript with some modifications and revised it according to the Reviewers’ comments. We provide a point-by-point response to the your comments.Please see the attachment.

Reviewer 2 Report
Dear Authors,
This is a thorough examination of the response of a planthopper to a generalist parasitoid. The experiments are well designed and replicated, analysis is good. Overall the manuscript is well written and the figures and tables sufficient to convey the results of the study.
There are some grammatical issues and points requiring clarification which are highlighted and commented upon in the manuscript.
A moderate revision should suffice to address the concerns found in the document.

Author Response
Dear Reviewer:
Thank you very much for your letter and the comments from the referees about our paper submitted toBiological traits of the pincer wasp Gonatopus flavifemur (Esaki et Hashimoto) associated with different stages of its host, the brown planthopper, Nilaparvata lugens (Stål) (ID: insects-770681). We have checked the manuscript with some modifications and revised it according to the Reviewers’ comments. We provide a point-by-point response to the your comments.Please see the attachment.

Reviewer 3 Report
Review of Biological traits of the pincer wasp Gonatopus flavifemur (Esaki et Hashimoto) associated with different stages of its host, the brown planthopper, Nilaparvata lugens (Stål) for Insects
The authors of the MS deals with a topic of some importance. The study showed highlight that Drinidae G. flavifemur can parasitize and feed upon all life stages of the brown planthopper with a marked difference in the parasitism rate, host-feeding rate and development of offspring. The pest species investigated is a herbivore of particular importance as it has a remarkable ability to spread quickly to its high biotic potential (high reproductivity, short life cycle, rapid dispersal capacity). The results obtained are interesting because the information acquired from the research is not available in the bibliography. In particular, the information reported becomes central in order to implement biological control since the brown planthopper is showing resistance to the
various insecticides used. Although the general interest in the research adopted in the MS, some details on the methodology used are missing as well as some clarifications are to be highlighted. The clarifications and improvements requested from the authors are reported in the general and specific comments of the revision.
General comment
Further but brief indications could be provided on the parasite such as those on the possibility of the herbivore to transmit viruses and also the relationship between the sexes of the species investigated, etc. Also, the authors could provide indications on the parasitization of the collected material. In assessing the effects of parasitization and predation of nymphs since it is not possible to distinguish the sex of the nymphs, the authors should indicate how they analyzed or explored differences in the sex ratio in potential nymphs in contact with a dryinid. The graphs (Fig. 2, 3, 4) proposed must be modified in tables. In this case it is necessary to add the column on the numbers of individuals for each treatment and not only that of the proportions obtained. Further information should be given on the dryinid number larvae that produced cysts and if were able to emerge from the nymph and reach the adult stage.
Specific comment
Line 36-37: “…male wasps do not parasitize and feed on their hosts…”. The sentence as written suggests that there may be species in which parasitoid males can parasitize host. Remove the sentence or edit the sentence;
Line 54: “however, the excessive and single use of the same chemical insecticide has caused the resistance of 54
BPHs to many insecticides”. Please add reference regarding resistence of herbivore. For example https://www.nature.com/articles/s41598-018-22906-5;
Line 76 - 77. “parasitism rate of 1.06–7.29% (unpublished data). These results are similar to those from previous studies. Gonatopus flavifemur is one of the most common parasitoids of BPHs in ...”. Which previous studies? The authors reported that similar investigated results are not yet published right? This way of presenting information needs to be changed;
Line 87: Please specify if TN1 is Taichung Native variety or other variety; Several studies show that this variety is indicated to carry out susceptibility studies to test for rice insect.;
Line 96: The authors need to provide more details of where the experimental tests were made. For example if the greenhouse is made of iron and glass how did they maintain a constant temperature of 27 ° C with an accuracy of ± 1 ° C?;
Line 99-100: The authors must report the reference bibliography or alternatively specify how they managed to identify the different stages of development of this Homoptera group. (Are they 5 or 6 young stages?);
Line 101: Can author specify if this Dryinidae species reproduction is bisexual and/or parthenogenetic?;
Line 107: It is not clear how the authors made sure of the host feeding and the superparasitization (quantity of parasitoid cysts?);
Line 120-121: “Please report the test used to if your data met the assumption of homogeneity of variances, only after this authors could use Tukey's honestly significant difference (HSD) post hoc test. Moreover, for percentage which transformation was adopted.
Line 128-129: Provide some more explanation of the "handling time" in the test adopted;
Line 139-140: F distribution has two sets of degrees of freedom: “numerator” and “denominator please report them and also the number of individuals.
For the figures, see the general comments;
In the y-axis of Fig. 5 Gonatopus flacifemur in Gonatopus flavifemur
Author Response

(The authors gave the same response as above.)

Round 2
Reviewer 3 Report
I congratulate you on the result achieved in the experimentation and having contributed to providing interesting information on this species and its type of parasitism.
Interested to see your further developments I send you best regards
Author Response
Dear Reviewer:
Thank you very much for your suggestions and comments on our manuscript, we will continue to conduct the further studies in the future based on present results.
Yours sincerely
Jiachun He& Qiang Fu